# Nurse-led telerehabilitation intervention to improve stroke efficacy: Protocol for a pilot randomized feasibility trial

Stephanopoulos Kofi Junior Osei[1]*, Emmanuella Adomako – Bempah[2], Adelaide Appiah Yeboah[3], Lawrence Akuamoah Owiredu[2], Lillian Akorfa Ohene[1]

1 School of Nursing & Midwifery, University of Ghana, Accra, Ghana, 2 37 Military Hospital, Accra, Ghana, 3 University of Ghana Hospital, Accra, Ghana

* stephanpaosei@gmail.com

## Abstract

### Background

The prevalence of stroke continues to rise in low-middle income countries. The continual rise in stroke cases and increasing prevalence on post-acute needs represent a crucial call for increased accessibility and utilization of rehabilitation services.

### Aim

The primary objective of the study is to test the feasibility of a nurse-led telerehabilitation intervention in improving self-efficacy among stroke survivors. The findings of the trial are intended for use in a future larger study.

### Methods

Participants would be recruited at the University of Ghana Hospital and randomized into an intervention group and a control group. Participants aged $\geq$ 18 years, diagnosed of stroke at most 12months prior the recruitment and requiring moderate level of assistance would be considered for eligibility. Participants in the intervention group will receive individualized and comprehensive nurse-led rehabilitation therapies in physical, emotional, cognitive and nursing education domains for 6 months, in addition to treatment as usual (TAU). The control group will only receive treatment as usual. Follow-up evaluations will occur immediately, 30 days and 90 days after the intervention.

### Discussion

Providing stroke rehabilitation services in low-resource settings presents a significant challenge due to limited infrastructure and a lack of trained healthcare professionals. The current study has the potential of contributing to the growing body of evidence on the impact of telerehabilitation services in mitigating these challenges in low-resource settings.

**Data Availability Statement:** No datasets were generated or analysed during the current study. All relevant data from this study will be made available upon study completion.

**Funding:** This study was made possible by a grant from the World Innovation Summit for Health (WISH) a member of the Qatar Foundation. Funding was received by SKJO. Website: https://2022.wish.org.qa/ The funding body did not contribute the study design.

**Competing interests:** The authors have declared that no competing interests exist.

**Abbreviations:** IRB, Institutional Review Board; LMICs, Low and Middle Income Setting; PC, Personal Computer; TAU, Treatment; WHO, World Health Organization.

## Trial registration

PACTR202210685104862, Pan African Clinical Trial Registry.

## Introduction

In low- and middle-income settings, stroke remains one of the major causes of disability and mortality [1–3]. Patients who suffer a stroke may experience variable degrees of impairment and, in more serious situations, death. The aftermath of a stroke depends on several factors, including patient characteristics, emergency health-seeking behaviour, the nature of the stroke, and post-stroke sequelae [4–6]. It is crucial for stroke survivors with impairments to undergo rehabilitation as part of the recovery process [7].

According to Knect et al. [8], rehabilitation after stroke plays a critical role in improving functional, and cognitive outcomes. Other studies have shown that providing emotional and psychological support for the patient and their caregivers during the rehabilitation process can greatly contribute to overall success [9, 10].

Traditionally, stroke rehabilitation begins after the patient has been physiologically stabilized and is still on hospital admission [11]. This initial rehabilitation phase is primarily focused on improving physical functioning and facilitating the patient's transition back to their home. This is followed by continuous rehabilitation, which typically takes place between 8 weeks and 6 months after the onset of the stroke. During continuous rehabilitation, patients make practical adjustments in their homes and try to navigate their natural environment with minimal assistance [12].

Despite the enormous impact of rehabilitation on recovery and life adjustment post-stroke, the delivery of equitable and affordable access to rehabilitation remains a challenge globally [13]. This challenge is more prominent in low and middle income countries (LMICs) due to limited rehabilitation infrastructure and inadequate number of trained health workers [14]. In Ghana, access to rehabilitation especially during the continued phase and post-hospital discharge is met with major challenges, including the high cost of available services, limited dedicated stroke rehabilitation services and units, socio-cultural barriers, long waiting times, poor education on rehabilitation and poor communication with healthcare providers [14, 15]. As a result, persons recovering from stroke may have several unmet rehabilitative needs which remains unresolved, and adversely impact the recovery process [14].

These challenges exist partly due to the nature of conventional stroke rehabilitation during the continuous phase. Patients who are undergoing post-discharge rehabilitation mostly have to present themselves at the health facility, receive planned therapy and travel back to their homes [15]. This approach however is wrought with issues such as long travel distances and financial burden which often deter the stoke survivors from returning to use available services inf any. During the COVID-19 pandemic, such travels were virtually impossible in some settings creating a greater need to consider other innovative measures of continuing rehabilitative support even at a distance [16].

Telerehabilitation is becoming a highly considered approach to delivering stroke rehabilitation services. The approach has yielded positive outcomes, including improved stroke self-efficacy, motor functioning, and emotional well-being [17–19]. Telerehabilitation–an emerging means of delivering rehabilitation services remotely using information technology- is projected to be a tremendous solution to the current challenges associated with rehabilitation accessibility. Additionally, a few studies focusing stroke telerehabilitation in Ghana has found the approach to be feasible, acceptable among patients and clinicians and appropriate for addressing challenges such as barriers to physiotherapy [20, 21].

With a limited number of stroke specialists in the country, stroke nurses and community nurses in Ghana have to take a prominent role in supporting stroke rehabilitation [22]. As indicated earlier, the disabling effects of stroke put patients in a critical situation for increased rehabilitation support. Nurses are uniquely placed to support these patients after discharge. According to Kirvevold [12] nurses' roles in post-discharge stroke rehabilitation includes but are not limited to coordinating treatment and care, providing assistance to survivors and family, providing informational support, consolation, monitoring condition progress, and supporting the resumption of roles and activities. These are all major roles that are crucial for improving patient outcomes and meeting the rehabilitation needs of stroke survivors and caregivers [23]. However, similar to conventional stroke rehabilitation, nurses' involvement in stroke telerehabilitation is limited, particularly in LMICs [24]. Inadequate specialized nurses, limited policies supporting their role in stroke rehabilitation, and a lack of recognition by other healthcare professionals and caregivers contribute to limitations of nurses' involvement and practice in stroke rehabilitation [25]. Regardless, post-discharge stroke rehabilitation aligns with the mandate of nursing to promote recovery which places nurses at the core of the process. Despite this assertion, the critical role of nurses in post-discharge stroke rehabilitation especially in LMICs has not been well articulated in existing literature.

It is against this background that an innovative programme of care was developed. The pilot phase of the study seeks to understand the feasibility of a nurse-led telerehabilitation underpinned by the need to support stroke survivors in ADLs performance accomplishments.

## Conceptual framework

The development of the intervention is guided by the self-efficacy model, which posits that an individual's belief in their ability to perform to achieve certain goals–in this case, activities of daily living–plays a significant role in their motivation and behavioral change [26]. The model suggests that such belief is influenced by a number of sources including mastery experiences, vicarious experiences, social persuasion, and emotional state [27]. In the context of stroke rehabilitations, improving self-efficacy could help stroke survivors better engage in rehabilitation activities and ultimately improve their functional outcomes [28].

The nurse-led telerehabilitation intervention in this study is designed to improve the self-efficacy of stroke survivors by targeting the factors that influence self-efficacy belief. The intervention aims to provide stroke survivors with opportunities for mastery experiences, as well as vicarious experiences through modelling and guidance through achieving daily activities goals. Nursing support would also provide feedback and emotional support to further boost the stroke survivors' self-efficacy.

The study objective is to assess the feasibility of a nurse-led holistic telerehabilitation among Ghanaian stroke survivors. The primary objectives of the trial are as follows:

1. Assess the recruitment and retention rates of participants.

2. Assess the user satisfaction of the nurse-led telerehabilitation intervention.

3. Examine the factors influencing user satisfaction of the nurse-led telerehabilitation intervention

4. A secondary objective of the trial is to measure whether a nurse-led telerehabilitation would improve self-efficacy of stroke survivors.

## Methods and analysis

### Study design

The trial is a parallel-group (intervention and control groups) randomized controlled study with 1:1 allocation to assess feasibility of a nurse-led telerehabilitation intervention administered remotely at the homes of Ghanaian stroke survivors. Post intervention process evaluation will employ a mixed method approach to exploring feasibility elements of the intervention.

### Study setting

Study participants would be recruited from the University of Ghana Hospital in Accra, Ghana. The facility has a dedicated stroke outpatient clinic for clients and a physiotherapy unit. We have already engaged with the chief nurse officer, lead physiotherapist, and head of the stroke clinic of the hospital, who are key to the success of the participant recruitment process.

### Eligibility criteria

Study participants will be considered for inclusion if they meet the following criteria:(1) is 18 years or above (2) diagnosed of a stroke with an onset of 1 week to 12 months prior the intervention (3) has FIM score of 55–100 (Level 3 moderate assistance) (4) total NIHSS score of 5–15 (moderate stroke). However, study participants would be excluded subject to these criteria: (1) has no access to a smartphone and standard internet connectivity (1.0 mgbs) (2) is unavailable for follow-up interviews (3) has a cognitive impairment (as documented in the patient's medical records) that makes it difficult to comprehend the information provided and limits capacity to provide consent.

### Recruitment and screening

Participants would be recruited from the medical department and the stroke clinic of the University of Ghana Hospital. Patients at the study setting would be approached onsite by recruitment nurses who are independent from the study investigators and provided comprehensive background information about the study, including its risks and benefits during first encounter as part of inviting them to consider participating. If they opt to participate, eligibility clinical screening based on aforementioned criteria will be completed by the recruiting nurse.

In this regard, the nurse will complete the FIM and NIHSS scores to ascertain eligibility. Informed consent would be obtained by researchers at this point (S1 Fig). This informed consent has been approved by the Noguchi Institutional Review Board 032/21-22.

### Randomization, allocation and blinding

After eligible participants have provided informed consent and undergone baseline screening, they would be randomly assigned to the intervention group (Nurse-led rehabilitation support + Treatment as usual) and control group (only treatment as usual).

Randomization would be conducted by an independent researcher who is not involved in the study. She will use a combination of block and stratified randomization to assign 40 participants to either the intervention group or the control group. Prior to the assignment, participants would be stratified based on the type of stroke (hemorrhagic vs ischemic). Within each stratum, participants will be assigned using block randomization with block size of four. Additionally, the independent researcher would generate random sequence for the block order with an online random sequence generator (sealedenvelope.com). The sequence numbers would be written and placed in opaque envelopes by the independent researcher. Immediately

a participant signs the consent form, the principal investigator (SKJO) would inform to sequentially open the envelope in the corresponding block and assign participant to the group indicated in the envelope.

Participants will not be blinded to the assigned group. Also, It will not be possible to blind the interventionists (stroke nurses and other health professionals) providing the intervention in this study. Essentially both parties (participants and interventionists would be unmasked. However, the assessors of study outcomes would be blinded to the allocation of participants. Also, assessors gathering baseline data on self-efficacy and any other data after group allocation would be blinded to the allocation status of participants.

## Intervention

Eligible participants would randomly be assigned to either the treatment or control group. Participants in control group would receive treatment as usual (TAU) which comprise of two weekly in-person physiotherapy sessions and two sessions with a medical doctor at the University of Ghana Hospital Participants in the control group would receive upper and lower limb physical rehabilitation and undergo medical review (medication review and symptom management) with medical doctors.

Participants in the intervention group would receive comprehensive nursing neurorehabilitation support in addition to existing service at the hospital (treatment as usual). The components of the intervention are guided by relevant literature, established guidelines [29–31] and have been reviewed by a panel of experts including two neuroscience nurses, two community health nurses, three neurologists, one stroke specialist, one physiotherapist and a stroke survivor. Patients would undergo assessment and receive various therapies in appropriate domains for 6months. A follow-up evaluation would be performed at the end, 30- and 90-days after the intervention (S2 Fig). The intervention would involve the following components;

**Initial and ongoing patient assessment.** Participants would undergo various assessments on motor function, cognitive capacity, anxiety and depressive levels and nutritional status utilizing evidenced-based assessment tools. The initial assessment would span in the first week of the intervention. Based on the problems and needs identified during the assessment, participants would receive a bundle of interventions in the domains of cognitive rehabilitation, physical rehabilitation, emotional support, and stroke education. These interventions would be provided twice a week in 1hr-to-1hr 30mins per session via video calls on WhatsApp or phone calls. These calls would be scheduled weekly with the participants after needs and availability assessment to improve response. Participants with close-by caregivers would be asked to provide the contacts of their caregivers as back-up. A participant in the intervention may have 192 sessions maximum or 48 sessions minimum in the span of the intervention.

**Cognitive rehabilitation domain.** This domain is mainly focuses on supporting participants to establish a daily routine, improving memory and attention and mitigating spatial neglect [32]. We would guide participants in clearly structuring their daily activities based on their preferences and setting priorities [9]. Participants also be guided in visual scanning activities, attention training exercises and utilizing memory aids [33].

**Physical rehabilitation domain.** The aim of physical rehabilitation will be the recovery of the patient's motor functioning impacted by stroke [8]. Participants would undergo stroke-specific impairment assessment using the Fugl-Meyer Assessment (FMA) [34]. Based on identified needs the participant would undergo graded physical therapy including passive and activities range of motion exercises, constraint-induced exercises, and stress ball usage with the support of their caregivers. Caregivers would be provided education on how to support

participants through the therapy, and handle equipment such as stress ball. Additionally, participants and their caregivers would be educated on proper use of assistive devices for gait and mobility training.

**Emotional support.** This domain is geared towards assessing and supporting the emotional status of participants and their caregivers. We would provide counselling sessions to help identify the extent of loss which greatly influences the emotional status and recovery of participants [35]. Additionally, study participants would be guided in recognizing their own emotions and identifying personalized but effective coping strategies [9].

**Stroke education.** Education lasting from 30 to 60mins based on participant's and caregiver's learning needs would be provided for participants and their relatives per session. The target of the information provided would focus on improving the participant's understanding of the disease condition, modifiable risk factors of recurrence, pain management, preventing pressure ulcers, oral and catheter care, continence management, the importance of following professional dietary recommendations, and healthy nutrition in the recovery process [36].

See (Table 1) for details regarding the intervention presented based on the TiDieR checklist.

## Outcomes and outcome measures

**Primary outcomes.** Based on the CONSORT checklist [37] the pilot study is primarily aimed at assessing the preliminary feasibility and acceptability of the intervention in the Ghanaian context.

*Feasibility*. Feasibility would be measured by retention rate, rate of adverse events and recording the details of resources used in implementing the intervention. Additionally, duration of sessions and cost of internet for the research team and study participants would be captured.

*Participant satisfaction*. This will include the use of a 12-item Patient Satisfaction with Telehealth questionnaire to assess the level of satisfaction of participants on delivery of the intervention, time spent, quality of audio-visual channels involved and overall perceived benefits [38].

Factors influencing user satisfaction: Qualitative data of participant experiences and in-depth factors that influence their rate level of satisfaction would be captured in order to provide in-depth explanation of satisfaction levels. This would be guided by an interview probe designed for purpose.

**Secondary outcome.** *Self-efficacy*. The secondary and clinical outcome in focus would be Stroke Self-efficacy. The goal is assessed potential effectiveness of the intervention in improving self-efficacy among patients. This would be measured the Stroke Self Efficacy Questionnaire (SSEQ). The SSEQ is a 13-item scalar questionnaire that measures the stroke survivor's ability and confidence in performing tasks under self-care, mobility and effective coping. The scale is also vital for assessing the functional performance of the study participants after the intervention [39, 40]. This tool is most appropriate because it has been proven to have good face validity with a Cronbach Alpha of 0.90 suggesting good internal consistency and higher criterion validity when compared with other self-efficacy scales [41].

## Proposed sample size

We aim at recruiting 20 participants per group (total = 40 participants). The size however is guided by Julious' rule of thumb for 12 participants per group in pilot trial [42]. The potential attrition rate estimated is 20% per group. During the follow-up phase, qualitative data on

**Table 1. Details of intervention.**

| Component of the intervention | What (procedures) | Who | How (mode of delivery) | Where (location of intervention) | When (schedule), How much, dose, intensity | Tailoring |
|---|---|---|---|---|---|---|
| Initial comprehensive assessment | Patient's baseline assessment on:<br>• Motor functioning<br>• Cognitive capacity<br>• Levels of anxiety & depression | Stroke nurses, and physiotherapist and neuropsychologist as consultants. | • Individual one-on-one<br>• Calls via cell phones<br>• video calls | • Patient's home<br>• Rehab center | Only once as this serves as the baseline assessment. | All participants would undergo this initial assessment for identification problem. |
| Ongoing assessment | Identifying patient's<br>• needs<br>• priority areas<br>• progress and possible deterioration<br>• motor functioning<br>• cognitive capacity<br>• levels of anxiety and depression | Stroke nurses | • Individual one-on-one<br>• Calls via cell phones<br>• video calls | Patient's home | Bi-Weekly<br>Every other week<br>Monthly<br>As needed | • Individualized |
| Cognitive rehabilitation domain | • Guiding patients to establish a structured routine.<br>• Attention training exercises.<br>• learning the proper use of memory aids | Stroke nurses | • Individual one-on-one<br>• calls via cell phones.<br>• video calls | Patient's homes | • Every other week<br>• A total of 12 sessions throughout the project | Based on cognitive needs participants may be guided in using different strategies to:<br>• improve memory and attention.<br>• mitigate spatial neglect |
| Physical rehabilitation domain | • Passive and/or active range of motion exercises<br>• constraint induced therapy and stress-ball usage.<br>• Education on proper usage of assistive devices would also be provided | Stroke nurses, physiotherapist | • Individual one-on-one<br>• Calls via cell phones<br>• video call | Patient's homes | Twice a week. A maximum of 48 sessions throughout the project. | Physical domains sessions are tailored to the functional needs of participants. Based on physical needs, intervention may be guided in using different strategies to:<br>• Aid those who can walk and stand with aids.<br>• Those who cannot walk but stand with some gait issues.<br>• And those who cannot stand at all |
| Emotional support domain | • Understanding the extent of loss<br>• Recognizing emotions and mood changes<br>• Identifying personalized coping strategies | Stroke nurses | • Individual one-on-one<br>• calls via cell phones<br>• Video calls | Patient's homes | • Biweekly<br>• A maximum of 12 sessions throughout the project | Participants would have to the opportunity to identify individual coping strategies. |
| Stroke education domain | • modifiable risk factors<br>• pain management<br>• pressure ulcer prevention<br>• nutrition and continence. | Stroke nurses | • Individual one-on-one<br>• Calls via cell phones<br>• Video calls<br>• Group sessions | Patient's homes | Every three weeks during the intervention period. | Certain education topics would be provided on one-on-one and based on patient needs whereas topics on modifiable risk factors and nutrition may be provided in group sessions |

participant experiences through one-on-one interviews would be captured from 15 conveniently sampled participants from the intervention group. According to Hennick and Kaisser [43] a 15 sample is appropriate for reaching data saturation. An interview guide designed for purpose who be used to gather needed data.

## Data collection, entry and management

Data would be gathered from participants before the intervention but after recruitment and after the intervention. Baseline data would be gathered on two occasions prior to the intervention. Before randomization, data on participant demographic characteristics, patient history, functional status, psychological functioning and cognitive capacity would be gathered. Baseline data on patient's self-efficacy would be collected before the first session of the intervention via phone survey. The second phase of data collection would take place immediately after the end intervention capturing quantitative date on user satisfaction, qualitative data on experiences throughout the intervention only for participants in the intervention group and stroke-self efficacy levels for participants in both group. The third and final phase of data collection would be executed 4 weeks and 12 weeks after the intervention respectively capturing stroke efficacy levels in both groups. The data collection process would be undertaken by an independent assessor who are unaware of participant allocation. After completing each phase of data collection, the quantitative data would be entered into a spreadsheet using Microsoft Excel. The data would be processed for outliers and missing inputs. Qualitative interviews would be audiotaped, audios would be saved in a PC folder and later transcribed for analysis. The entered information and saved audios would be protected by a password only accessible by the research team.

## Data analysis

Quantitative data analysis would be conducted using SPSS v26. The socio-demographic characteristics of participants would be summarized in frequencies and percentages. Continuous data gathered would be scrutinized for normality and necessary transformation would be applied if needed. Continuous variables would be summarized using means and SD. Mann-Whitney test or its parametric equivalent (if possible) would be employed to analyses the difference stroke self-efficacy levels in both groups at same points of follow-up evaluation. A Friedman test or its parametric equivalent (if possible) would be employed to ascertain changes in stroke self-efficacy level over the time. Results would be classified as significant if $p < 0.05$ (two-tailed). All qualitative transcripts exported to NVivo version 10 would be analyzed by reading that data line by line and making noting down insights as meaning units while the reading is ongoing. The meaning units would be labeled with a code, this would be done repeatedly to capture all aspects of the content (data). The codes would then be categorized into sub-categories and broader categories for identification of themes [44]. Both quantitative and qualitative data would be compared to identify similarities and differences for the purpose of convergence.

## Ethics and dissemination

The current study is targeted at determining the feasibility of a nurse-led telerehabilitation intervention for stroke patients in a low-resource setting. The study will illuminate the level of feasibility and effectiveness of this approach in settings similar to Ghana. Additionally, the study is crucial in further development and designing for similar interventions aimed at increasing accessibility to stroke rehabilitation using digitalisation. The study has ethical clearance from the Noguchi Memorial Institute of Research–IRB. Additionally, the trial has been registered in a public domain (PACTR202210685104862, Pan African Clinical Trial Registry). In the case of protocol modifications, both the IRB and trial registry would be informed using appropriate channels.

The findings of the study would be disseminated through peer-reviewed journals, nursing online and offline communities and conferences. Various key stakeholders including the Noguchi Memorial Institute of Research, the Ministry of Health Ghana and the University of Ghana Hospital (Legon) would be provided feedback and possibly recommendations based on

the findings associated with the proposed study. De-identified dataset at individual participant-level will be disseminated through peer review journals.

## Possible risks and mitigation

There are minimal perceived risks to study participants involved in the intervention. However, all participants would undergo periodic risk assessments to unearth potential risks of adverse outcomes. The assessment would be carried out in their homes by the research investigators. Additionally, participants and their caregivers would be provided emergency dial lines and the contacts of the researchers in case of an adversity. In case of an adverse event such as falls or emotional collapse, the researcher (and stroke nurse/ other rehabilitation health worker) working with the patient at moment would immediately assess the situation, notify emergency support if participant is in immediate danger (e.g. ambulance service, national suicide hotlines and participant's emergency contact) for immediate assistance and continue to engage with the participant and/or their caregivers until help arrives. If the participant is not in immediate danger, they would be referred to the University of Ghana Hospital for further assistance. All adverse events related to the intervention would be well-documented, reported to clinicians in charge and reviewed. Additionally, the research team has liaised with community health nurses to provide assessment and further recommendation for patients in their homes.

## Discussion

Accessibility to stroke rehabilitation in low-resource settings remains a prevalent challenge due to limited infrastructure and inadequately trained healthcare professionals [13, 14]. In Ghana, the challenges are exacerbated especially after hospital discharge by barriers including limited communication from health workers, socio-cultural barriers, inadequate rehabilitation education, high cost of available services and long waiting times [15].

There is evidence that nurses–the largest health professional group in Ghana, are uniquely placed to support rehabilitation services and increase accessibility [12].

The current study is deemed to reveal the effects of a nurse-led telerehabilitation intervention on improving the self-efficacy of stroke survivors in a low-resource setting. The intervention is holistic and draws underpinning insights from relevant literature, a multidisciplinary team and patients.

Although the approach is therapy-based, the primary goal is to offer individualized and family-centred rehabilitation support coordinated by nurses. Langhorne, Bernhardt & Kwakkel [45] asserts that multidisciplinary involvement, individualizing rehabilitation intervention and increasing intensity is crucial to the improvement of patient outcomes.

The study would also shed light on the feasibility of this emerging approach in delivering stroke rehabilitation services in a low-resource setting like Ghana. Additionally, the study would contribute to the existing literature on the potential impact of telehealth on healthcare delivery services in Ghana and other LMICs.

## Limitations

The study is being conducted on a feasibility scale and involves a small size which may not be sufficient to detect significant differences between study groups albeit the findings can offer useful insight into telerehabilitation services for stroke survivors. There is a also notable difference in the dose and type of the intervention for the control and interventional group which could impact the validity of outcomes. Additionally, the short-term follow-up periods of the study limit the ability to explore long-term effects of the intervention.

## Conclusion

The implementation of a nurse-led telerehabilitation intervention for stroke survivors offers a potential contribution to mitigating the major challenges that threaten the accessibility of stroke rehabilitation services after acute management and post-hospital discharge.

## Supporting information

**S1 Fig. Enrolment, interventions and assessment schedule.**
(DOCX)

**S2 Fig. Workflow chart.**
(DOC)

**S1 File. SPIRIT checklist.**
(DOCX)

**S2 File. CONSORT checklist.**
(DOCX)

## Acknowledgments

The authors would like to acknowledge the immense support offered by Miriam Iddrisu, Joyce Pwavra and Samuel Adjorlolo (PhD) all of the University of Ghana School of Nursing and Midwifery. We are also grateful to Dr Ram Hariharan, Dr Siva Nair, Nurse Hellen Oteng and Jonathan Bayuo (PhD) for their brilliant comments on this manuscript.

## Author Contributions

**Conceptualization:** Stephanopoulos Kofi Junior Osei, Emmanuella Adomako – Bempah, Adelaide Appiah Yeboah, Lawrence Akuamoah Owiredu, Lillian Akorfa Ohene.

**Funding acquisition:** Stephanopoulos Kofi Junior Osei, Emmanuella Adomako – Bempah, Adelaide Appiah Yeboah, Lawrence Akuamoah Owiredu.

**Investigation:** Stephanopoulos Kofi Junior Osei.

**Methodology:** Stephanopoulos Kofi Junior Osei, Adelaide Appiah Yeboah, Lawrence Akuamoah Owiredu.

**Project administration:** Stephanopoulos Kofi Junior Osei.

**Supervision:** Lillian Akorfa Ohene.

**Writing – original draft:** Stephanopoulos Kofi Junior Osei.

**Writing – review & editing:** Stephanopoulos Kofi Junior Osei, Lillian Akorfa Ohene.

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
