## [Decision Letter · Decision Letter 0]

14 Feb 2023

PONE-D-23-00839Nurse-Led Telerehabilitation Intervention to Improve Stroke Efficacy: Protocol for a pilot randomized feasibility trialPLOS ONE

Dear Dr. Osei,

Thank you for submitting your manuscript to PLOS ONE. After careful consideration, we feel that it has merit but does not fully meet PLOS ONE’s publication criteria as it currently stands. Therefore, we invite you to submit a revised version of the manuscript that addresses the points raised during the review process.

We look forward to receiving your revised manuscript.

Kind regards,

Doris YP Leung

Academic Editor

PLOS ONE

Journal Requirements:

a) If there are ethical or legal restrictions on sharing a de-identified data set, please explain them in detail (e.g., data contain potentially sensitive information, data are owned by a third-party organization, etc.) and who has imposed them (e.g., an ethics committee). Please also provide contact information for a data access committee, ethics committee, or other institutional body to which data requests may be sent. Please note that authors, including Corresponding Authors, are not permitted to be the sole point of contact for data requests.

b) If there are no restrictions, please provide the minimal anonymized data set necessary to replicate your study findings as either Supporting Information files or to a stable, public repository and provide us with the relevant URLs, DOIs, or accession numbers. For a list of acceptable repositories, please see http://journals.plos.org/plosone/s/data-availability#loc-recommended-repositories.

Additional Editor Comments:

The topic is important and the research study is very relevant for developing countries. The manuscript is in general in good quality with a detail description of the intervention. However, I agree with the reviewers that more justifications on the use of nurse-led rehabilitation program in the background and detail descriptions of the methodology of the study protocol are needed. Besides, I would also recommend to report recruitment rate, which is an important measure for feasibility study.

Reviewers' comments:

Reviewer's Responses to Questions

**Comments to the Author**

1. Does the manuscript provide a valid rationale for the proposed study, with clearly identified and justified research questions?

Reviewer #1: Partly

Reviewer #2: Yes

2. Is the protocol technically sound and planned in a manner that will lead to a meaningful outcome and allow testing the stated hypotheses?

Reviewer #1: Partly

Reviewer #2: Partly

3. Is the methodology feasible and described in sufficient detail to allow the work to be replicable?

Reviewer #1: No

Reviewer #2: Yes

4. Have the authors described where all data underlying the findings will be made available when the study is complete?

Reviewer #1: Yes

Reviewer #2: Yes

5. Is the manuscript presented in an intelligible fashion and written in standard English?

Reviewer #1: Yes

Reviewer #2: Yes

6. Review Comments to the Author

You may also provide optional suggestions and comments to authors that they might find helpful in planning their study.

Reviewer #1: Thank you for inviting me to review this feasibility protocol. The paper is well written and easy-to-follow. I would have some suggestions for the authors.

1. line 52-53, "rehabilitation mainly targets the recovery of motor and cognitive functions". This is a bold statement even the authors have cited a reference. Literature has shown that there were many programs focusing on not only physical, but psychosocial needs of stroke patients. This should not be a gap for your study.

2. line 66, place full name of LMIC

3. line 99-101, why nurses' involvement in stroke telerehabilitation is limited?

4. line 153, who are the researchers? How to randomize? Who randomize the participants? Blinding?

5. line 157-8, what would physiotherapy and medical doctors do in the control group?

6. line 191, this is the first time I can see a caregiver in the manuscript. Will all the participants be accompanied by their caregivers? If not, will the author record it down in demographic data? In addition, will the participants do the physical exercise by themselves? Who will be given the eqipment, e.g. stress ball? If there is problem, like fall, who will be responsible? what will be done?

7. During emotional assessment, if the patient has sudden emotional collapse (or want commit suicide), what will the nurse do? any precautionary measures?

8. How long is the stroke education?

9. How and when to call the participant? If the participants did not take the call, any back-up plans? why video call is needed in all domains?

10. How about if the patients do not have Wifi, will they be recruited?

11. Will the intervention group participant still need to return to hospital to receive usual care? If not, there is already an intervention difference between two groups. So are you testing the effect of telerehabilitation by nurse or the difference between hospital visit and telecare?

12. What is the reason for measuring self-efficacy? Does this intervention scheme guided by a conceptual framework?

13. What is the reason to recruit 15 sample for qualitative interview? How is the interview, group or individual? Any framework guided?

14. line 259, who are the investigators? blinded?

15. line 261, what is the design of the qualitative interview? what methods are you going to use? how about triangulation, credibility?

Reviewer #2: Thank you for submitting this interesting protocol. The main consideration is the study design; see comments within the appropriate section below;

Abstract

Remove words in bracket after feasibility [Within the aim]

Methods – Study type. More succinct overview

Background

Whilst there is key information presented, relating to the study, it would be beneficial to keep this background more succinct and focused on the study (nurse lead telerehab). From the background it remains unclear what telerehabilitation is and whether previous research has considered the nurse role within stroke and telerehabilitation, whether telerehabilitation is used in other areas within the country and any challenges identified relating to the telerehabilitation prior to designing this study.

Aim – Remove the word ‘preliminary’. Suggest placing user satisfaction as an objective rather than in the main aim

Objective 1; Add recruitment i.e. assess the recruitment and retention

Objective 3; ‘Participant experience’; Unclear how this is different from objective 2

Place ‘ A secondary objective’ within the objectives, as objective 4

Recruitment and screening;

Line 146- It is unclear who will first approach the patients? Also will that person have already pre identified potential participants and how?

Line 150; Formatting

Line 156; Randomisation process is unclear

Line 158 – Is this control intervention at home? This is unclear

Whilst the focus is to assess compared to a control, there are significant difference in the two i.e. dose of intervention and type of intervention therefore it might be better to focus solely on the telerehabilitation at this stage.

Line 169; Should this be a sub title?

Table 1; How and where not needed as this is the same throughout; this could be addressed above the table

Line 218; Participant satisfaction; Name this questionnaire

Line 221 – From the outset the method should be clear, that this is an interview

Move paragraph at 240 into the main methods

Line 265 – The authors should consider if the statistical tests are appropriate given this is a feasibility study with small numbers.

Line 280 – Risks with remote delivery should be outlined. Also consider what happens to participants should they no longer have remote access

7. PLOS authors have the option to publish the peer review history of their article (what does this mean?). If published, this will include your full peer review and any attached files.

Reviewer #1: **Yes: **Wong Arkers Kwan Ching

Reviewer #2: No

---

## [Author Response · Author response to Decision Letter 0]

30 Mar 2023

Stephanopoulos K.J. Osei,

University of Ghana School of Nursing & Midwifery,

Accra, Ghana.

24th March, 2023.

Doris YP Leung,

Academic Editor,

PLOS ONE.

Response to Editor & Reviewers’ Comments: PONE-D-23-00839

Nurse-Led Telerehabilitation Intervention to Improve Stroke Efficacy: Protocol for a pilot randomized feasibility trial

Dear Dr Leung,

We are grateful for the opportunity to submit a revised draft of our manuscript “Nurse-Led Telerehabilitation Intervention to Improve Stroke Efficacy: Protocol for a pilot randomized feasibility trial” to PLOS ONE. We appreciate the time and effort that you and the reviewers have dedicated to providing brilliant feedback on the manuscript. We have incorporated the insights provided in our revised manuscript and below is a point-by-point response to the reviewers’ comments and concerns.

Editor’s comments Response

Please ensure that your manuscript meets PLOS ONE's style requirements, including those for file naming. All formatting and file naming requirement has been followed

We note that you have indicated that data from this study are available upon request. PLOS only allows data to be available upon request if there are legal or ethical restrictions on sharing data publicly. For more information on unacceptable data access restrictions. A revision on data sharing has been indicated on line 356 of the manuscript. Essentially, de-identified data would published.

Your ethics statement should only appear in the Methods section of your manuscript. Ethics statement has been moved to the methods section on line 330

Review #1 Authors’ Response

1. line 52-53, "rehabilitation mainly targets the recovery of motor and cognitive functions". This is a bold statement even the authors have cited a reference. Literature has shown that there were many programs focusing on not only physical, but psychosocial needs of stroke patients. This should not be a gap for your study. The statement has been removed and modified as seen on line 58

2. line 66, place full name of LMIC Thank you for pointing this out. We agree with this comment. Therefore, have affected the change and put down the full meaning of LMIC as Low and middle-income countries. Line 71

3. line 99-101, why nurses' involvement in stroke telerehabilitation is limited? We agree and have, accordingly, put together evidence on why nurses' involvement in stroke telerehabilitation is limited on line 110 with references made to authors with papers in that field of study. Mentioned evidence includes inadequate specialized rehabilitation nurses, lack of policies supporting the role of nurses in stroke rehabilitation 

4. line 153, who are the researchers? How to randomize? Who randomize the participants? Blinding?

 A major revision has been made on line 186

5. line 157-8, what would physiotherapy and medical doctors do in the control group? This has been clarified on line 208.

6. line 191, this is the first time I can see a caregiver in the manuscript. Will all the participants be accompanied by their caregivers? If not, will the author record it down in demographic data? In addition, will the participants do the physical exercise by themselves? Who will be given the equipment, e.g. stress ball? If there is problem, like fall, who will be responsible? what will be done?

 Thank you for pointing this out. The role of caregivers has been pointed out in line 231. Most of the patients being recruited at the moment have dedicated caregivers who would support the role. Details on adverse events such as falls have been discussed in line 364.

7. During emotional assessment, if the patient has sudden emotional collapse (or want commit suicide), what will the nurse do? any precautionary measures? Thank you for drawing our attention to such a valid issue. We have explained in line 287 the nurses’ interventions as immediate assessment of the situation, notifying emergency support (national suicide hotlines, ambulance services) if participant proves to be in immediate danger for immediate assistance. However, if the participant is not in any immediate danger, he/she would be referred to the University of Ghana Hospital for further assistance.

In regards to the precautionary measures, we have addressed this in line 337

8. How long is the stroke education? We appreciate this comment and have accordingly placed this information in line 256 where we explained that stroke education will last 30 to 60 minutes based on the participant’s and caregiver’s learning needs

9. How and when to call the participant? If the participants did not take the call, any back-up plans? why video call is needed in all domains?

 We have explained how calls would be scheduled and also highlighted that participant’s caregivers would provide their contacts as back on Line 229

10. How about if the patients do not have Wifi, will they be recruited?

 Patients with no internet connection would be excluded from the study. Most Ghanaian smart phone users use mobile data instead of wifi plans. Line 170

11. Will the intervention group participant still need to return to hospital to receive usual care? If not, there is already an intervention difference between two groups. So are you testing the effect of telerehabilitation by nurse or the difference between hospital visit and telecare?

 Thank you for your comment. Yes, participants in the intervention group will still need to return to the hospital to receive usual care as we hope to test the effect of a nurse-led telerehabilitation not the difference between hospital visit and telecare. This has been highlighted in line 207

12. What is the reason for measuring self-efficacy? Does this intervention scheme guided by a conceptual framework?

 The conceptual framework guiding the study has been described on line 127

13. What is the reason to recruit 15 sample for qualitative interview? How is the interview, group or individual? Any framework guided?

 The qualitative phase of the project is

necessary to explore the impact of

intervention among study participants.

The interviews will adopt an individual

one-on-one approach, which will be

guided by interview guide on the domains of function of the stroke survivor and factors influencing their satisfaction rating. The changes could be located on line 295

14. line 259, who are the investigators? blinded? The investigators have been indicated in line 319. Blinding of participants in the interventionists and participants have been indicated under the randomization section on line 200

15. line 261, what is the design of the qualitative interview? what methods are you going to use? how about triangulation, credibility? The qualitative interviews will adopt an

individual one-on-one approach, which

will be guided by semi-structured

interview guide. The qualitative phase will ensure triangulation and increase credibility of the study. Kindly find changes in line 342

Reviewer #2 Authors’ Response

Abstract

Remove words in bracket after feasibility [Within the aim] Thank you for your suggestion. We have addressed it in line 18 

 Methods – Study type. More succinct overview

 Agree. We have accordingly, revised the methods to provide a more succinct overview in line 22-30

Aim – Remove the word ‘preliminary’. Suggest placing user satisfaction as an objective rather than in the main aim We appreciate this remark and have removed the word Line 141 

Background

Whilst there is key information presented, relating to the study, it would be beneficial to keep this background more succinct and focused on the study (nurse lead telerehab). From the background it remains unclear what telerehabilitation is and whether previous research has considered the nurse role within stroke and telerehabilitation, whether telerehabilitation is used in other areas within the country and any challenges identified relating to the telerehabilitation prior to designing this study. In line 91, a definition of what telerehabilitation has been provided. Also the use of telerehabilitation is stroke care has been indicated in line 96, previous studies that have highlighted the role of telerehabilitation have been indicated. The role of nurse with rehabilitation has been indicated on line 107 and the barriers associated with nurses being involved in conventional stroke rehabilitation and telerehabilitation has been indicated on the line 110.

Objective 1; Add recruitment i.e. assess the recruitment and retention We have made the modification in Line 144

Objective 3; ‘Participant experience’; Unclear how this is different from objective 2 We have made the modification in Line 148

Place ‘ A secondary objective’ within the objectives, as objective 4 We have made the changes in Line 150

Recruitment and screening;

Line 146- It is unclear who will first approach the patients? Also will that person have already pre identified potential participants and how? The clarification has been made in line 177. Essentially the participants would be approached and screened by recruitment nurses. 

Line 150; Formatting The formatting has been changed now on line 180

Line 156; Randomisation process is unclear Randomization process has been elaborated in lin 186. This includes the person in charge of the randomization, how sequence would be generated and how participants would be assigned to groups.

Line 158 – Is this control intervention at home? This is unclear The setting of the control intervention has been added on line 207.

Whilst the focus is to assess compared to a control, there are significant differences in the two i.e. dose of intervention and type of intervention therefore it might be better to focus solely on the telerehabilitation at this stage.

 We completely agree that there should be a match in the intensity and dose of therapies provided in the intervention group and that of the control group. Unfortunately, we cannot conduct this at the feasibility stage due to resource constraints. We would also like to point out that the comparison is only a secondary focus and in subsequent larger scale studies, the dose match would be highly considered. In the meantime, we will highlight this limitation in line 420

Line 169; Should this be a sub title?

 Yes, it is a subtitle under intervention and has been formatted as such.

Table 1; How and where not needed as this is the same throughout; this could be addressed above the table Some modifications has been made to the TiDier table as the where is not the same throughout

Line 218; Participant satisfaction; Name this questionnaire The questionnaire has been named as Participant Satisfaction with Telehealth in Line 276

Line 221 – From the outset the method should be clear, that this is an interview Thank you for pointing this out. The patient satisfaction would be measured quantitatively (user satisfaction questionnaire). Factors influencing the rated satisfaction would be explored using in-depth qualitative interviews. The clarification has been made on line 280.

Move paragraph at 240 into the main methods It has been moved to line 186. Thank you.

Line 265 (now on 279) – The authors should consider if the statistical tests are appropriate given this is a feasibility study with small numbers. Thank you for pointing this out. Since the sample size is small as indicated by you, the is a high possibility the assumption of repeated measures would not be met. We have made changes to the test in line 335.

Line 280 – Risks with remote delivery should be outlined. Also consider what happens to participants should they no longer have remote access Thank you for pointing this out. The possible risks of remote delivery of the intervention have been highlighted in line 364 (under possible risks and mitigation). Additionally, a comprehensive risk management protocol has been added in such situations. With regards to patient with no remote access, they would be excluded from the study as indicated in line 170. However, if you’re asking of patient who initially had internet to access the intervention but at some point no longer has such assess, they would be recorded as patients who left the program and the reasons would be recorded as indicated in the SPIRIT work flow.

Thank you for considering our manuscript.

Sincerely,

Stephanopoulos Osei (BSN, RN).

---

## [Decision Letter · Decision Letter 1]

4 May 2023

PONE-D-23-00839R1Nurse-Led Telerehabilitation Intervention to Improve Stroke Efficacy: Protocol for a pilot randomized feasibility trialPLOS ONE

Dear Dr. Osei,

Thank you for submitting your manuscript to PLOS ONE. After careful consideration, we feel that it has merit but does not fully meet PLOS ONE’s publication criteria as it currently stands. Therefore, we invite you to submit a revised version of the manuscript that addresses the points raised during the review process.

Data collection procedure needs further elaboration.

We look forward to receiving your revised manuscript.

Kind regards,

Doris YP Leung

Academic Editor

PLOS ONE

Journal Requirements:

Additional Editor Comments:

There are still some points for clarification:

1. Line 312: Will the data collection at baseline be done before intervention or before randomization?

2. Line 322: SKJO will not be blinded of group allocation as SKJO will open the envelopes for randomization (line 199) and hence should not be involved in data collection after group allocation.

In addition, some typos were found:

3. Line 191: Typo “She will use of combination …”

4. Line 195: ‘researcher’ instead of ‘research’

Reviewers' comments:

Reviewer's Responses to Questions

**Comments to the Author**

1. Does the manuscript provide a valid rationale for the proposed study, with clearly identified and justified research questions?

Reviewer #1: Yes

2. Is the protocol technically sound and planned in a manner that will lead to a meaningful outcome and allow testing the stated hypotheses?

Reviewer #1: Yes

3. Is the methodology feasible and described in sufficient detail to allow the work to be replicable?

Reviewer #1: Yes

4. Have the authors described where all data underlying the findings will be made available when the study is complete?

Reviewer #1: Yes

5. Is the manuscript presented in an intelligible fashion and written in standard English?

Reviewer #1: Yes

6. Review Comments to the Author

You may also provide optional suggestions and comments to authors that they might find helpful in planning their study.

Reviewer #1: Thanks for the authors' effort in addressing my comments. My comments were well addressed. The revised version was much better now.

7. PLOS authors have the option to publish the peer review history of their article (what does this mean?). If published, this will include your full peer review and any attached files.

Reviewer #1: **Yes: **Wong Kwan Ching, Arkers

---

## [Author Response · Author response to Decision Letter 1]

7 May 2023

Response

Thank you for question. 

The correction has been effected in line 285. Essentially, baseline data would be gathered on 2 occasions and all two occasions have been clearly defined

Many thanks for pointing this out. 

SKJO and other investigators have been removed as assessors of patient outcomes in line 298 and 184. Instead, independent assessors who are blind to group allocation would be employed for this role.

Typo rectified in line 171

Type rectified in line 175

---

## [Editor Report · Decision Letter 2]

10 May 2023

PONE-D-23-00839R2Nurse-Led Telerehabilitation Intervention to Improve Stroke Efficacy: Protocol for a pilot randomized feasibility trialPLOS ONE

Dear Dr. Osei,

Thank you for submitting your manuscript to PLOS ONE. After careful consideration, we feel that it has merit but does not fully meet PLOS ONE’s publication criteria as it currently stands. Therefore, we invite you to submit a revised version of the manuscript that addresses the points raised during the review process.

The data collection procedure is still unclear, which needs clarifications.

We look forward to receiving your revised manuscript.

Kind regards,

Doris YP Leung

Academic Editor

PLOS ONE

Journal Requirements:

Additional Editor Comments:

This sentence is unclear: whether the baseline data on self-efficacy be collected, after or before the first session of the intervention? Also, it is clear that this baseline data will be collected after randomization. Will the assessor be blinded or able to be blinded to the group allocation?

Line 291: Baseline data on patient’s self-efficacy would be collected after before the first session of the intervention.

---

## [Author Response · Author response to Decision Letter 2]

10 May 2023

That has been clarified. The data on patient’s self-efficacy would be gathered prior to the first session. Also in line 185, all assessors gathering data post randomization would be blinded to patient allocation.

---

## [Editor Report · Decision Letter 3]

17 May 2023

Nurse-Led Telerehabilitation Intervention to Improve Stroke Efficacy: Protocol for a pilot randomized feasibility trial

PONE-D-23-00839R3

Dear Dr. Osei,

We’re pleased to inform you that your manuscript has been judged scientifically suitable for publication and will be formally accepted for publication once it meets all outstanding technical requirements.

Kind regards,

Doris YP Leung

Academic Editor

PLOS ONE

Additional Editor Comments (optional):

All the comments have been well addressed.
---

## [Editor Report · Acceptance letter]

24 May 2023

PONE-D-23-00839R3 

 Nurse-Led Telerehabilitation Intervention to Improve Stroke Efficacy: Protocol for a pilot randomized feasibility trial 

Dear Dr. Osei:

I'm pleased to inform you that your manuscript has been deemed suitable for publication in PLOS ONE. Congratulations! Your manuscript is now with our production department. 

Kind regards, 

on behalf of

Dr. Doris YP Leung 

Academic Editor

PLOS ONE